# Aberrant Epigenomic Modulation of Glucocorticoid Receptor Gene (NR3C1) in Early Life Stress and Major Depressive Disorder Correlation: Systematic Review and Quantitative Evidence Synthesis

**DOI:** 10.3390/ijerph16214280

**Published:** 2019-11-04

**Authors:** Laurens Holmes, Emily Shutman, Chinacherem Chinaka, Kerti Deepika, Lavisha Pelaez, Kirk W. Dabney

**Affiliations:** 1Nemours Healthcare System for Children, Translational Health Disparities Science Research Program, Wilmington, DE 19803, USA; Eshutman@haverford.edu (E.S.); Chinacherem.Chinaka@nemours.org (C.C.); Keerti.D@yahoo.com (K.D.); Lavisha.Pelaez@nemours.org (L.P.); Kirk.Dabney@nemours.org (K.W.D.); 2Biological Sciences Department, University of Delaware, Newark, DE 19716, USA; 3College of Population Health, Thomas Jefferson University, Philadelphia, PA 19107, USA; 4Biological Sciences Department, Haverford College, Haverford, PA 19041, USA; 5Department of public health, Eastern Virginia Medical School, Norfolk, VA 23507, USA; 6Community Environmental Health Department, Old Dominion University, Norfolk, VA 23507, USA; 7Sidney Kimmel Medical School, Thomas Jefferson University, Philadelphia, PA 19107, USA

**Keywords:** early life stress (ELS), glucocorticoid receptor gene (NR3C1), major depressive disorder (MDD), DNA methylation (mDNA), aberrant epigenomic modulation

## Abstract

Early life stress (ELS) induced by psychological trauma, child maltreatment, maternal separation, and domestic violence predisposes to psycho-behavioral pathologies during adulthood, namely major depressive disorder (MDD), anxiety, and bipolar affective disorder. While environmental data are available in illustrating this association, data remain to be established on the epigenomic underpinning of the nexus between ELS and MDD predisposition. Specifically, despite the observed aberrant epigenomic modulation of the NR3C1, a glucocorticoid receptor gene, in early social adversity and social threats in animal and human models, reliable scientific data for intervention mapping in reducing social adversity and improving human health is required. We sought to synthesize the findings of studies evaluating (a) epigenomic modulations, mainly DNA methylation resulting in MDD following ELS, (b) epigenomic modifications associated with ELS, and (c) epigenomic alterations associated with MDD. A systematic review and quantitative evidence synthesis (QES) were utilized with the random effect meta-analytic procedure. The search strategy involved both the PubMed and hand search of relevant references. Of the 1534 studies identified through electronic search, 592 studies were screened, 11 met the eligibility criteria for inclusion in the QES, and 5 examined ELS and MDD; 4 studies assessed epigenomic modulation and ELS, while 2 studies examined epigenomic modulations and MDD. The dense DNA methylation of the 1F exon of the NR3C1, implying the hypermethylated region of the glucocorticoid receptor gene, was observed in the nexus between ELS and MDD, common effect size (CES) = 14.96, 95%CI, 10.06–19.85. With respect to epigenomic modulation associated with child ELS, hypermethylation was observed, CES = 23.2%, 95%CI, 8.00–38.48. In addition, marginal epigenomic alteration was indicated in MDD, where hypermethylation was associated with increased risk of MDD, CES = 2.12%, 95%CI, −0.63–4.86. Substantial evidence supports the implication of NR3C1 and environmental interaction, mainly DNA methylation, in the predisposition to MDD following ELS. This QES further supports aberrant epigenomic modulation identified in ELS as well as major depressive episodes involving dysfunctional glucocorticoid-mediated negative feedback as a result of allostatic overload. These findings recommend prospective investigation of social adversity and its predisposition to the MDD epidemic via aberrant epigenomic modulation. Such data will facilitate early intervention mapping in reducing MDD in the United States population.

## 1. Introduction

### 1.1. Early Life Stress and Biological Underpinnings of Epigenomic Modulation 

Early life stress (ELS) such as perinatal, intrapartum, prenatal, and maternal stress has been indicated to influence physical and psychological health in adulthood. The observed connection is explained by social signal transduction, which results in molecular level events such as gene transcription inhibition, impaired mRNA translation, and gene expression, influencing physiologic function and disease development. For example, maternal stress and isolation in rodents and humans have been implicated in glucocorticoid gene receptor dysregulation, which is involved in allostatic stress response and pro-inflammatory cytokine elaboration [1]. The epigenomic mechanistic process upon which inappropriate protein in synthesized involves the methylation of DNA in the promoter region of the gene or acetylation phosphorylation of the histone protein at the amino-terminal region (lysine) [2].

Studies have illustrated the mediating biologic consequences of maternal deprivation and lack of care of offspring in animal models via gene and environment interaction, as indicated by aberrant epigenomic modulation of the candidate genes involved in these conditions [3,4,5,6]. Similarly, low SES(Socio-economic status), which reflects income inequalities, implies variabilities in access to social, educational, health, and other resources, resulting in status shame, anxiety, depression, self-harm, and other psychopathologies [7,8,9,10]. Data on social hierarchies driven by high SES observed privileged access to social determinants of health (SDH) and basic life resources such as water, food, and shelter as an appropriate living condition. The SES, which correlated with anxiety in most cases as observed in a capitalistic society, reflects social adversity as exposure function of childhood physical and emotional neglect, restricted control over decision-making and life choices, as well as marginalized social interaction and decreased trust and confidence with societal members [11]. Epidemiologic and clinical data indicate hypercortisolemia and increased pro-inflammatory cytokines in low social ranked individuals as a marker of social gradient in health [7]. Available animal and human model studies observed DNA hypermethylation of the CpG regions of the genes such as NR3C1 involved in allostatic stress response. However, it is not fully understood if the neural activities involving the hypothalamus pituitary adrenal (HPA) axis and the cortisol, mediated negative feedback mechanism induced by social signal transduction via social isolation, low SES, unstable social status, and ELS, such as child abuse and neglect, regulated solely NR3C1.

The conserved transcriptional response to adversity (CTRA) gene expression has been observed in low socioeconomic status, chronic stress, posttraumatic stress, and other life stressors [1]. Reliable findings in humans and animal models have linked the CTRA genes with the upregulation of pro-inflammatory response as well as inhibition of the anti-inflammatory mediators such as interferon-gamma (IFN-γ) and IgG. The mechanical process involves the leukocytes and the mediation through the innate immune mechanism, namely the macrophages as well as the specific immune response through the activated lymphocytes, namely the CD4 cells. Specifically, the methylation process of this gene (CTRA) involves the binding to the gene promoter region being the 5’cytosine with the methyl group resulting in 5’methylcytosine due to the excessive availability of the DNA methyltransferase enzyme and the subsequent inhibition of the transcription factors negatively impacting the CTRA expression, abnormal protein synthesis, and cellular function dysregulation. Specifically, the CTRA gene expression involving the leukocytes results in the pro-inflammatory elaboration, namely IL-2 and IL-6 [1]. The expression of the CTRA gene due to chronic stress increases the immune system response with pro-inflammatory cytokines and inhibition of IFN-γ elaboration required in response to a virally infected cells, as well as a decreased response of IgG which is a specific therapeutic antibody needed for bacterial pathogens destruction via immune complex formation (antigen and antibody interaction). 

Cells have a robust system for gene expression that is highly regulated by various signaling molecules and RNAs. Gene expression is controlled by specific transcription factors that bind to CpG islands in the promoter region to initiate transcription of a particular mRNA sequence, which is then translated into a specific protein sequence. This process is highly regulated by various developmental, physiologic, pathological, and environmental cues that ensure that the necessary protein is produced in response to specific stimuli. While epigenomic modulation does not directly involve the DNA sequence, genomic mutations can change the base sequence in a portion of the genome. If located in the protein-coding portion of a gene or if it results in a frameshift mutation, this will often deleteriously affect the resultant protein sequence.

In contrast, other mechanisms do not change gene sequence but affect gene expression by conferring long-term programming to genes. These are called epigenetic changes and are defined by “a long-term change in gene function that persists even when the initial trigger is long gone and does not involve a change in gene sequence or structure [12].” Some definitions mandate heritability in dividing somatic cells while other definitions simply require a change in gene expression. 

Epigenetic changes can be conferred in multiple ways, all of which affect the accessibility of the DNA to the transcription machinery. Inaccessible genes are relatively silenced, while accessible genes can be actively transcribed. One of the most predominant ways in which this silencing occurs is through modifications of the chromatin—the protein-based structure around which the DNA is wrapped. The chromatin protein is composed of nucleosome “beads,” each of which is composed of eight histone proteins. The N-terminal tails of these histones are frequently modified by methylation, acetylation, phosphorylation, and ubiquitination [13]. Typically, histones are positively charged, which allows favorable electrostatic interactions to occur between the DNA backbone and the histone proteins, enabling the DNA to wrap tightly around nucleosomes. The aforementioned modifications neutralize the charge on the tail, weakening the interaction between the genome and the nucleosome core. Weak interactions between the genome and histones reduce chromatin density, enabling greater penetrance of transcription factors, which, in turn, increases transcription. In contrast, demethylation and deacetylation reduce gene accessibility and decrease transcription. The mechanisms of these modifications are not fully understood, but histone demethylase enzymes have been implicated in the mechanism of histone demethylation [14].

An alternative mechanism of epigenetic regulation involves the DNA molecule itself. In vertebrates, the identity of a cell is regulated by methylation at the 5’ position of the cytosine ring in the CG dinucleotide sequence. When this methylation occurs in the promoter region, transcription factors are unable to bind to initiate transcription of a particular gene. This principle confers cell identity in that differing patterns of methylation between different cell types allows for specific activation or silencing of specific genes. This enables different cells of an organism to produce different protein products and execute different functions [15]. While most research on genome-based methylation is focused in the promoter region, there is evidence that methylation in other gene elements plays a role in regulating gene function. For example, decreased DNA methylation in gene bodies is associated with increased chromatin accessibility, which is likely due to a disturbance in the favorable electrostatic interaction between the DNA and the chromatin [12].

Furthermore, some evidence suggests that small interfering RNAs (siRNAs) have epigenetic implications. Available data suggest siRNAs have a role in both DNA methylation and histone modifications. A study observed that siRNAs impair the function of DNMT1, 3a, and 3b, which are key DNA methylation enzymes [16]. In addition, miRNAs may regulate chromatin structure by impairing the function of a histone modifier enzyme called histone deacetylase 4 [17].

### 1.2. Epigenetic Implications of Childhood Trauma

The aforementioned epigenetic mechanisms all have the effect of increasing or decreasing accessibility to the genome, impacting the level of gene expression by way of mRNA production. These epigenetic mechanisms are highly impacted by environmental stimuli and can be observed in response to chronic stress or trauma [18]. These epigenetic effects may be able to explain the correlation between childhood trauma or early life adversity (ELA) and health outcomes in adults at a biological level. Studies pertaining to the epigenetic effects of childhood trauma and subsequent health outcomes often reference the NR3C1 gene—a gene that codes for the protein of the glucocorticoid receptor. This receptor is the site at which cortisol and other glucocorticoids bind. When the receptor binds to glucocorticoids, its primary mechanism of action is the regulation of gene transcription. The receptor–cortisol complex effectively works to decrease inflammation in tissues by either upregulating the expression of anti-inflammatory proteins in the nucleus or by repressing the expression of pro-inflammatory proteins in the cytosol [19]. It executes the latter by preventing the transport of transcription factors from the cytosol to the nucleus. In effect, the reduction in inflammation is a negative feedback mechanism that terminates the stress response after the end of a threat [20]. There is evidence that decreased NR3C1 expression in the hypothalamic–pituitary–adrenal (HPA) center is often observed in suicide victims with a history of childhood abuse.

The activity of this gene can be affected by methylation at the promoter region of the gene. Specifically, previous study identified that DNA methylation at a CG dinucleotide in exon 5, which is located in the promoter, caused decreased mRNA production. This was determined by using a DNA microchip assay to ascertain the level of hybridized NR3C1 mRNA in the cytoplasm. This study indicated lower levels of the mRNA than expected, demonstrating methylation at the promoter that impaired transcription. Previous studies indicated that low levels of NR3C1 mRNA were observed in individuals with co-occurring child abuse and suicide, schizophrenia, and depression [13]. This relationship can be explained by the methylation at exon 5, which caused low levels of NR3C1 expression. Low levels of glucocorticoid receptor actually induced increased activity in the hypothalamic–pituitary–adrenal (HPA) center, which results in altered stress responses, likely due to an increase in inflammation of the region. Ultimately, the altered stress response impairs the efficacy of the negative feedback loop, mirroring a situation in which the individual has an increased perception of being in a threatening situation. Specifically, this renders individuals less equipped to handle daily stressors and is implicated in serious psychological conditions.

Similar correlations between childhood trauma and disadvantageous health outcomes have been observed in reference to the FKBP5 gene. This gene codes for a protein that is an important functional regulator of the glucocorticoid receptor complex. Childhood trauma, compounded with a specific polymorphism called rs1360780, caused a 12.3% decrease in DNA demethylation on intron 7 of the gene—a site near the transcription start site [21]. This demethylation increases FKBP5 transcription which ultimately decreases glucocorticoid receptor sensitivity, leading to a prolongation of stress hormone system activation following exposure to stress, and is linked to many stress-related psychiatric disorders, namely predisposition to PTSD and depression. 

### 1.3. Structural Inequity in Childhood Trauma

While the biological mechanisms of DNA (de)methylation can happen to any individual who experiences childhood trauma, it is important to note the inequity in which different populations experience trauma. Childhood trauma—stemming from neglect, abuse, and violence—are more prevalent in populations that are already structurally disadvantaged. Accumulated structural disadvantage, as manifested in limited access to healthy foods, housing insecurity, and internalized societal racism, can exacerbate the biological effects of trauma in marginalized populations. For this reason, an efficacious study will examine the correlation among childhood trauma, gene expression, and adult health outcomes in reference to different subpopulations. 

### 1.4. Current Study Objective

The current study aimed to: (a) assess evidence in published literature on early life stress and major depressive disorders in adulthood, as well as correlate the DNA methylation profile on these conditions, (b) quantitatively synthesize the results of these published literature using random-effect meta-analysis procedure, and (c) examine heterogeneity of the studies to determine the reliability of the combined or common effect sizes (CES).

## 2. Experimental Section

### 2.1. Design

This study involves a systematic review and applied meta-analysis termed quantitative evidence synthesis (QES). The utilization of QES ensures a quantitative systematic review with common effect size in providing a summary estimate on the epigenomic modulation involved in the glucocorticoid gene (NR3C1) receptor, thus allowing for scientific statement of dense DNA methylation of NR3C1 in ELS and major depressive disorder (MDD).

The search included identifying relevant literature, selecting articles based on a set of predetermined inclusion criteria, and performing study quality assessment. The qualitative synthesis of selected studies included data abstraction and summary. The QES included data extraction from eligible published literature, the pool assessment estimation, test for heterogeneity, and the creation of forest plots.

The overarching objectives of QES are to (1) minimize random error and (2) marginalize measurement errors which have a huge effect on the point estimate by down-drifting away from the null. Because all studies have measurement errors, and some studies have more measurement errors than others, QES assesses the differences between studies that are due to measurement errors. Additionally, because studies in medicine and public health are often conducted with small samples, such samples have increased random errors. The QES, which is a method of summarizing the effect across studies, increases the study or sample size and, therefore, minimizes random error and enhances generalizability of findings. Furthermore, QES integrates results across studies to identify patterns and to some extent establish causation. In effect, the overall relevance of QES is to generate scientific data that is accumulative and reliable in improving health or other conditions upon which it is applied. 

The methodology used in QES [22] differs from that of traditional meta-analysis. While meta-analysis utilizes fixed and random effect methods, QES only employs random effect method and examines heterogeneity after and not before the pool estimates. The fixed effect method is only applicable to QES when the combined studies or publications are from multicenter trial where the study protocol is identical. However, when studies are combined from different settings, observation and measurement errors induce significant variability in the observed estimates, limiting such combination without adjusting for between studies variability. The random effect method compensates for the between studies variability, hence its unique application in QES. 

Scientific endeavor makes sense of the accumulating literature in medicine and public health given the confounding and contradicting results. QES reflects such attempts at study integration for public health and clinical decision-making. 

A unique feature of QES is temporality, in which findings in QES accumulate with time. For example, if QES was performed on the implication of epigenetic as a consequence of early childhood trauma and adult-onset MDD, this study must identify the time of conduct and continue to add findings and reanalyze the data for contrasting or negative findings with time. Subsequently, the emergence of new data on epigenetic modulation (physical, in utero, social, endocrine, neurobiologic, environment) informs the results of QES by shifting the direction of the previously observed data, implying the change in the results of QES, thus moving evidence in a different direction. 

Science and scientific endeavors are not static but dynamic (continually shifting and modifying as new evidence emerges), implying the emergence of new data creating a shift in evidence. The scientific community cannot wait until evidence accumulates to such a point that no further addition is required with respect to evidence discovery in order to initiate an intervention. Consequently, QES can inform and generate knowledge required in risk characterization (specific risk characterization), subclinical and clinical disease process, disease prognosis, as well as control and disease prevention at the population level. 

### 2.2. Search Engine and Strategies 

We conducted the online database search in June 2018 of MEDLINE via PubMed. Search terms were created based on medical subject headings (MeSH) and terms used in epigenetic literature reviews of ELS and MDD in order to maximize sensitivity:
(gene expression OR DNA methylation OR histone acetylation OR microRNA OR gene transcription OR siRNA OR mRNA OR histone methylation OR epigenotype AND early childhood trauma) OR (gene expression OR DNA methylation OR histone acetylation OR microRNA OR gene transcription OR siRNA OR mRNA OR histone methylation OR epigenotype AND schizophrenia) OR (gene expression OR DNA methylation OR histone acetylation OR microRNA OR gene transcription OR siRNA OR mRNA OR histone methylation OR epigenotype AND neuroendocrine development (cortisol, pituitary hypothalamus, HPA axis, vasopressin, corticotropin hormone, CRF, ERA, glucocorticoid receptor gene, (NR3C1), brain derived neurotropic factor (BDNF), glucocorticoid receptor (GR)), GABA aminobutyric acid AND major depressive disorders AND human)

Additionally, we performed hand searches through reference lists of relevant articles. Such articles were identified in advance based upon their investigation of epigenetic alterations and MDD. 

### 2.3. Eligibility Criteria 

One author (ES) screened abstracts for inclusion in the full-text evaluation. Eligible articles had to meet the following criteria: (a) study published in English from 1 January 2000, to 1 June 2018; (b) study investigates epigenetic changes and ELS as well as MDD; (c) study with a well-defined outcome, namely MDD and ELS; and (d) study that contains quantitative data, such as the parameter values (odds ratio, risk ratio, relative risk). Studies with loss to follow-up >25% or sample size smaller than 10 were excluded to lessen the likelihood of selection bias and sparse data bias, respectively (QES). Inclusion criteria were developed in order to maximize incorporation of any potentially useful findings while limiting inclusion of irrelevant data. When a study was identified that alluded to the existence of quantitative data but did not disclose any of it in the article itself, we contacted the authors via e-mail in an effort to obtain additional data. 

### 2.4. Data Extraction Reliability

Two authors (ES and KD) independently read the full texts (kappa = 9.3, indicative of excellent inter-rater agreement). Discrepancies in agreement were resolved through discussion between the study investigators. The final list of studies included in the qualitative and quantitative syntheses was the product of this discussion following the initial independent review. Studies included in the qualitative synthesis had to meet the inclusion criteria above, implying studies involving epigenetic modulation measured by DNA methylation and its association with ELS and MDD.

### 2.5. Study Variables 

The study variables included early life stress (ELS) and major depressive disorders (MDD) as the response. Epigenetic alterations as well as neuroendocrine markers of neurobehavioral dysfunctions were the independent variables, namely glucocorticoid receptor gene (NR3C1) and the DNA methylation profile. Cellular function commences with the central dogma of molecular biology: replication as the origin of cellular function, transcription, RNA translation, and protein synthesis. 

### 2.6. Data Collection and Processing

We collected data from all eligible studies based on the outcome variables and specific research questions. For the qualitative synthesis, all available data concerning epigenetic markers and MDD as well as ELS were obtained. For the QES, we abstracted data on DNA methylation percentage from ELS versus non-ELS for evidence of dense DNA methylation in ELS. We also applied the same approach in MDD with DNA methylation percentage as well as relative point estimate. When data were not available on the measure of the outcome, we estimated that the absence or presence of epigenetic changes and the outcome of MDD or ELS and manually estimated the precision measure of the parameter estimate using 95% confidence interval (CI).

### 2.7. Study Quality Assessment 

Two authors (ES and KD) assessed the eligible studies’ quality based upon study design, sampling techniques, clarity of aims or purposes, and adequacy of statistical analysis. Studies were also assessed for any confounding factors that might have influenced the outcomes and any potential bias, including selection, information, and misclassification bias. This study quality assessment technique was adequately comparable to the preferred method of reporting for systematic reviews [PRISMA statement] [23].

### 2.8. Statistical Analysis: QES 

The QES analysis that involved the pooled estimate for the CESs was performed prior to the heterogeneity test. In addition, we created a template to transform the proportion of epigenetic modulation and MDD cases into percentage, standard error, and 95% confidence intervals to enable the application of meta-analytic command using STATA, namely *metan percent lower CI upper CI*, label (namevar = study) *random*.

To test the hypothesis with respect to the implication of dense DNA methylation in ELS or MDD as well as MDD due to ELS, we used the random effect analysis of Dersimonian–Laird. The Dersimonian–Laird method was applicable given significant studies’ heterogeneity, while the fixed method was employed given absence of heterogeneity (homogenous studies). This procedure, namely random effect method, examines the between studies effect as well as the effect sizes of the combined studies weighing each study for their contributions to the overall sample size involved in the pool estimate. In addition, the study precision was measured by the 95% confidence interval. The effects size was estimated by testing the null hypothesis that the CES = 0, based on the standardized normal [z statistic].

We performed heterogeneity test to determine variability among studies overall and within each outcome category. The heterogeneity test was performed using Q = (1/variance_i) * (effect_i − effect_pool) ^2^. The variance was estimated using: variance_i = (upper limit − lower limit) / (2*z) ^2^. The test of heterogeneity reflects the variation in the effects size (ES) that is attributable to the differences between studies effect sizes.

The significance level was 5% (0.05 Type I error tolerance) and all tests were two-tailed. All analyses were performed using STATA 15.0 (StataCorp, College Station, TX, USA).

## 3. Results and Discussion

### 3.1. Results

These data represent a review of published literature on epigenomic programming modulations with respect to early childhood trauma or early life stress (ELS) as social adversity and the correlation with behavioral pathology, namely major depressive disorders (MDD). Eleven studies were identified that met the criteria for quantitative evidence synthesis (QES) as an applied meta-analysis and provided substantial information in transforming the point estimate to a CES measure as point estimate or perimeter value, as well as the precision measure, such as the confidence intervals. Of these eleven studies, five studies with DNA methylation involving ELS and MDD correlation were incorporated into the QES. Regarding early life stress, there were four studies that met the inclusion criteria, while two studies on major depressive disorder and DNA methylation met the criteria as well. Of the 1534 studies identified through the search of literature, 592 were screened; with 386 studies involving ELS and MDD correlation as well as MDD methylation, while 206 studies met the inclusion criteria for ELS DNA methylation (Figure 1).

#### 3.1.1. NR3C1 DNA Methylation in Early Life Stress and Major Depressive Disorder Correlation 

Table 1 demonstrates the studies with DNA methylation (mDNA) in childhood adversity and adult major depressive disorder (MDD) correlation. The sample size for the combined studies in the QES assessment was n = 531, while the study size was k = 5. The epigenomic mechanistic process used in all the studies was the DNA methylation with the epigenomic modulation involving the transcription region of the gene, namely the promoter (or enhancer for the transcription mechanism). These studies examined the mDNA of the glucocorticoid receptor gene, NR3C1, on the 1F exon region, the promoter sites of NR3C1. The DNA methylation was observed to increase in all the studies, implying dense DNA methylation in the correlation between ELS and MDD. 

Figure 2 presents the effect sizes of the individual studies on DNA methylation of the NR3C1 gene in the association between early childhood adversity as ELS and MDD. The combination of these individual study effect sizes resulted in the pool estimate of the overall effect size in this association. The individual studies incorporated in this QES all observed increased DNA methylation ranging from 1.10% to 37.0%. The combined or pool summary estimates indicated a significant 14.96% dense or increased methylation associated with epigenomic modulations implicated in the association between ELS and MDD, with common effect size (CES) = 14.96%, 95% CI = 10.06–19.85. The heterogeneity (I^2^) indicates significant differences in the individual effect sizes that generated the CES, I^2^, χ^2^ (4) = 1038.7, *p* < 0.001. The test of the variance from zero in the CES indicated a significant difference, implying CES > 0, z = 5.75, *p* < 0.001.

The subgroup DNA methylation indicated increased or dense DNA methylation following childhood trauma as social adversity (SA). The DNA methylation profile for SA indicated a CES = 10.45, 95%CI 5.57–15.34. Child sexual abuse was associated with 5.4% hypermethylation, while ELS due to parental post-traumatic stress disorder was associated with hypermethylation of 37% (Figure 3). 

#### 3.1.2. NR3CI DNA Methylation in ELS

Table 2 presents the studies that examined the DNA methylation profile in childhood adversity. There were four studies with quantitative methylation profile that constituted the QES assessment. These studies examined different CpG’s within the 1F exon, namely CpG 3, 5, 6, and 7. Overall, the sample size for this study was 361. The epigenomic mechanism of modulation was DNA methylation involving the promoter region where transcription factors are influenced. 

Figure 4 illustrates the forest plot of the methylation differential profile, comparing mDNA mean index in subjects with and without ELS. Relative to subjects without ELS, there was a 23% increased DNA methylation profile among those with ELS as childhood adversity, CES = 23.2%, 95% CI = 8.00–38.48. These studies were significantly heterogeneous with respect to the individual effect sizes, implying the between studies variance, I^2^, χ^2^ (3) = 4217, *p* < 0.001. The test of the variance from zero in the CES indicated a significant difference, implying CES > 0, z = 2.99, *p* = 0.003.

#### 3.1.3. NR3CI DNA Methylation and MDD

Table 3 exhibits the studies with DNA methylation associated with MDD. The overall sample size of the two studies was 642, which is a reasonable study size to assess the methylation differences with respect to hyper or hypomethylation. The epigenomic mechanistic process involved DNA methylation at the promoter or enhancer region of NR3C1 at 1F exon. The two studies in the QES did not observe a substantially dense DNA methylation.

Figure 5 shows the forest plot of the combined effect size that reflects the contribution of individual studies. A slightly increased methylation of the candidate gene namely NR3C1 in MDD, implying NR3C1 downregulation was observed. There was a 2.1% increased methylation among those with MDD compared to those without the disease, CES = 2.12%, 95% CI, −0.63–4.86. These studies were significantly heterogeneous implying, the between studies variance, I^2^, χ^2^ (1) = 75.08, *p* < 0.001. The test of the variance from zero in the common CES indicated no significant difference, implying CES ≤ 0, z = 1.5, *p* = 0.13.

### 3.2. Discussion

The need for a scientific statement and evidence-based understanding of the epigenomic modulation or programming in early childhood stress or adversity implying early life stress as well as the modification associated MDD and the correlation between ELS and MDD motivated the current systematic review and QES. This study assessed the epigenomic modulation in terms of DNA methylation mechanistic process in early life stress and MDD, as well as the relationship between ELS and MDD in terms of epigenomic aberrations induced by dense DNA methylation. There are a few relevant findings based on this QES. First, DNA methylation of the 1F exon on the NR3C1 gene, a glucocorticoid gene is associated with ELS and directly correlated with MDD in adulthood. Secondly, dense DNA methylation of the 1F exon of the NR3C1 gene is associated with increased risk of ELS. Thirdly, DNA hyper-methylation of the 1F exon of NR3C1 marginally increases the risk of MDD. 

We have demonstrated in this systematic review and QES that the DNA methylation of the glucocorticoid receptor gene (NR3C1) results in impaired gene expression and the abnormal protein synthesis, adversely affecting the cellular function of this receptor in response to cortisol and its biologic function in responding to stress and early life adversity, or early life stress. The observed hypermethylation of this receptor gene reflects impaired response resulting in chronic stress and subsequent mental illness given the pathway of cortisol response in MDD. The activation of the hypothalamus–pituitary–adrenal (HPA) axis by social signal transduction results in rapid response to stress and stressful environment and with inhibition of the transcription factor resulting in inactivation of the protein and receptor function. An example of social stressors as ELS has been illustrated in pre and postnatal stress and maternal care insufficiencies associated with behavioral pathologies [24,25,26]. Animal models with rodents demonstrated an inverse correlation between adequate postnatal maternal care and increased stress-reactivity, anxiety, and fear reaction [5]. 

In general, the main neural substrates involved in epigenetic modulation of ELS and maternal care include HPA, amygdala, medial prefrontal cortex, and hippocampal region of the brain. The observed phenotypes in the aberrant epigenomic modulation in ELS are explained by the NR3C1 gene downregulation and decreased expression, a glucocorticoid receptor (GR) gene. The animal model in this context observed the dense methylation of the CpG in the promoter region of NR3C1, increased activity of the HPA axis and elevated serum glucocorticoids level [5,6,27]. Additionally, animal model reveals increased stress reactivity, anxiety, impaired social interaction, and depressive episodes associated with maternal separation. The observed manifestations correlated with genome-wide aberrant DNA methylation and brain neural pathway genes transcripts [28,29,30]. 

Prolonged activation of the HPA axis, altered gene transcription, and aberrant DNA methylation were observed in CNS and peripheral T lymphocytes among infants and juveniles deprived of maternal care [31,32,33,34,35,36]. Aberrant epigenomic modulation (mDNA) and hydroxymethyl cytosine at the brain-derived neurotrophic factor (BDNF) locus, a member of the nerve growth factor family of neutrophils, had been observed in adolescence and adulthood distress associated with maternal maltreatment during neonatal nursing of offspring. Other neural changes include hippocampus, amygdala, and medial prefrontal cortex [37,38,39,40]. Further studies observed increased expression of arginine vasopressin (Avp) and gene loci, a protein-coding gene for antidiuretic hormone (Vasopressin) associated with diabetes insipidus and DNA hypomethylation in paraventricular nucleus of the hippocampus in ELS, due to maternal separation. Similarly, the overexpression of proopiomelanocortin (pomc) gene loci, which is involved in the production of adrenocorticotropic hormone (ACTH) and binds to melanocortin 2 receptor (MC2R), stimulating cortisol release is observed in ELS associated with maternal separation [41]. The overexpression of NR3C1 inversely correlates with corticotrophin-releasing hormone (CRH) and transcription dysregulation in chronic stress induced by ELS [42]. 

Human models have clearly illustrated how prenatal, perinatal, infant, and adolescence social adversities exposure lead to aberrant epigenomic modulation that are transformational into adulthood. Whereas NR3C1 implicated in ELS in rodents, varieties of NR3CI sequences are observed as regulatory targets with the most consistent and common epigenomic modulation being hypermethylation at exon 1F of human NR3C1 gene in ELS or early life social adversity. The DNA sequence element that encodes methylation-sensitive binding site associated with neural activity-regulated transcription activator (NGFTA) is embedded in exon 1F region of the NR3C1. The hypermethylation of NR3C1 in ELS, implying prolonged stress that results in allostatic load, interrupts the glucocorticoids-mediated to the pituitary and hypothalamus, enhancing HPA axis persistent activation and the consequent psychopathology including chronic inflammation and major depressive disorders (MDD) [43]. A correlation between parental stress, such as domestic violence and intimate partner violence, and NR3C1 CpG hypermethylation had been observed. Likewise, child abuse and neglect had been shown to be associated with dense NR3C1 CpG methylation and the subsequent impaired NR3C1 transcription, leading to NR3C1 downregulation and decreased gene expression [44,45,46,47,48,49]. While NR3C1 hypermethylation results in NR3C1-reduced transcription, it is not clear if other pathways or epigenomic processes influence NR3C1 gene expression. However, noncoding RNA namely IncRNA, GASS, and miRNA have been shown to impair NR3C1 gene expression and protein activities [50,51].

Other epigenomic modulations associated with early life adversities in animal models have implicated similar genes in humans such as BDNF, CRH, PM2D1, and CHRBP [36,52]. Additionally, genes involved in early life adversities include MAOA, SLC17A3, MORC1, and FKBP5 [27,53,54,55,56,57]. Overall, child maltreatment implying abuse and neglect and parental care deprivation signal aberrant epigenomic modulation inducing psychopathology and chronic disease such as type II diabetes and hypertension in adolescence and adult life due to allostatic overload and marginalized glucocorticoid-mediated negative feedback to pituitary and hypothalamus.

We have demonstrated that ELS is associated with hypermethylation of glucocorticoid gene (NR3C1) receptor. These findings are supported by previous data including the individual studies in QES that observed hypermethylation of candidate gene expression with ELS. Specifically, the methylation of the promoter region of DNA in this context results in the impairment in RNA polymerase response with respect to transcription and translation and the inhibition of the transcription factor resulting in abnormal gene expression and irregular protein synthesis affecting the receptor functionality. Overall, the observed cellular function is due to this methylation as an effect of social transduction in ELS, such as isolation, sexual abuse, child neglect, violent environment experienced by children at very young age, and maternal isolation and separation. Similarly, ELS has been observed in previous studies to result in psychiatric condition, including though not limited to schizophrenia, anxiety, bipolar, personality disorders, and post-traumatic stress disorder (PTSD) [23]. To assess the molecular events associated with ELS and psychiatric conditions, those without ELS and MDD were examined. These studies clearly indicated the hypermethylation of glucocorticoid gene among those with depression who suffered ELS relative with those who did not. The finding in this systematic review is supported by previous data [23] in both animal and human models on epigenomic programming that indicated the epigenomic programming commencing from gametogenesis that are transgenerational and predispose to disease conditions, but reversible. Studies have also indicated that mental activity such as yoga, meditation, and social network improves individual genome as well as “community genome” to response to stress implying lower incidence and prevalence of MDD in such environments [45].

The evidence accumulated in this QES may be questionable due to the difficulties in determining the CpG sites with unknown biologic functions that may be incorporated into the methylation or the hypermethylation process while describing the findings in the original papers. Since the methylation state of differentiated tissues is considered to be highly specific, it is not very clear how methylation process in the circulating blood cells or saliva will provide information or data about changes in DNA from the less accessible tissue has the human brain and the autonomic nervous system, as well as the immune system cells. 

Despite the strength of these findings in correlating MDD with ELS and providing a scientific statement based on the NR3C1 DNA methylation, there are some limitations. This approach involves the synthesis and quantification of given studies in order to arrive at a CES as evidence in intervention or therapeutics. First, systematic review and QES as an applied meta-analysis is a retrospective design that is prone to information and selection biases. Additionally, some studies incorporated in this QES despite the similarity of the design, implying a comparable measure of effect or point estimate, some studies failed to control for confounding, since no single variable explains everything with respect to causation. In effect, the scientific evidence and statement generated by this QES are subject to biases and unmeasured as well as residual confounding. However, it is highly unlikely that the magnitude of evidence provided on hypermethylation of NR3CI gene in ELS and MD is driven slowly by these limitations.

## 4. Conclusions

In summary, dense DNA methylation of the glucocorticoid receptor gene (NR3C1) is associated with ELS that predisposes to MDD. Addressing maternal stress, separation, and child abuse and neglect has substantial potential in lowering the incidence and prevalence of behavioral pathologies, namely major depressive disorder and bipolar affective disorders in all human societies. 

Whereas these findings implicate epigenomic modulation of NR3CI in ELS, as well as the nexus between ELS and MDD, caution is required in the interpretation and the application of these QES in intervention mapping, given the reversible nature of epigenomic modulations, as well as the timing of specimen collection in causal inference. Overall, this data suggests effective policy implementation and evaluation in ensuring that all children regardless of birth accidence are provided with adequate environment that supports proper growth and development, whereby, protective against subjective and objective adversity. 

## Figures and Tables

**Figure 1 ijerph-16-04280-f001:**
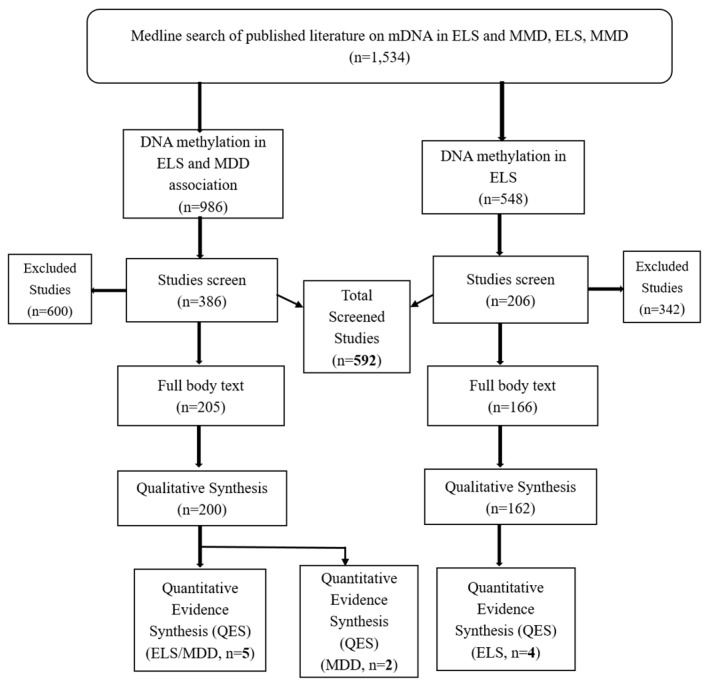
DNA methylation (mDNA) of candidate gene (1F exon of NR3C1 (glucocorticoid receptor) gene) associated with early life stress (ELS) and major depressive disorder (MDD).

**Figure 2 ijerph-16-04280-f002:**
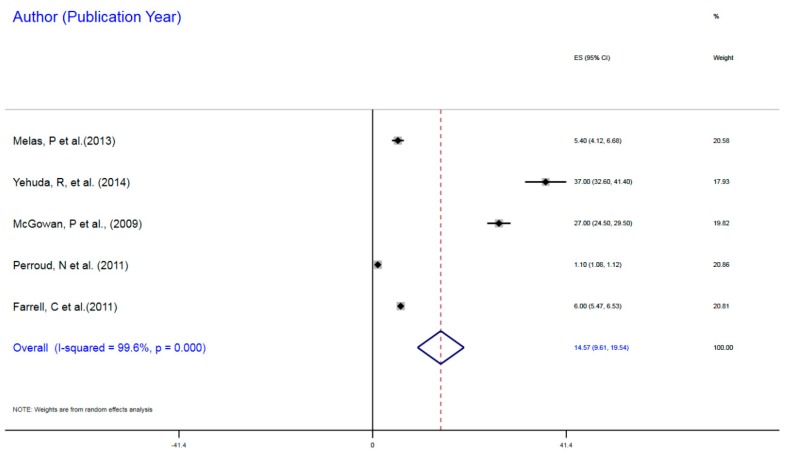
DNA Methylation of NR3C1 (glucocorticoid receptor gene) in Social Adversity as Early Life Stress and Major Depressive Disorder Correlation.

**Figure 3 ijerph-16-04280-f003:**
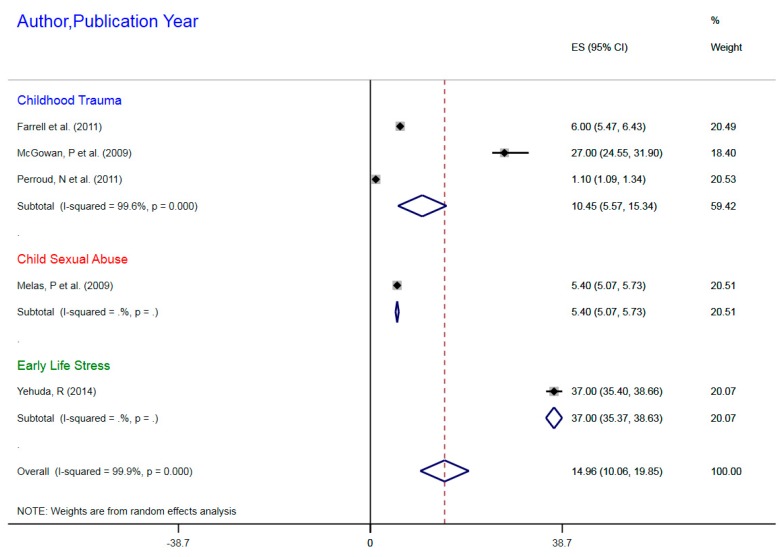
Meta-regression of NR3C1 (Glucocorticoid receptor) gene DNA Methylation in Social Adversity and Major Depressive Disorder Correlation.

**Figure 4 ijerph-16-04280-f004:**
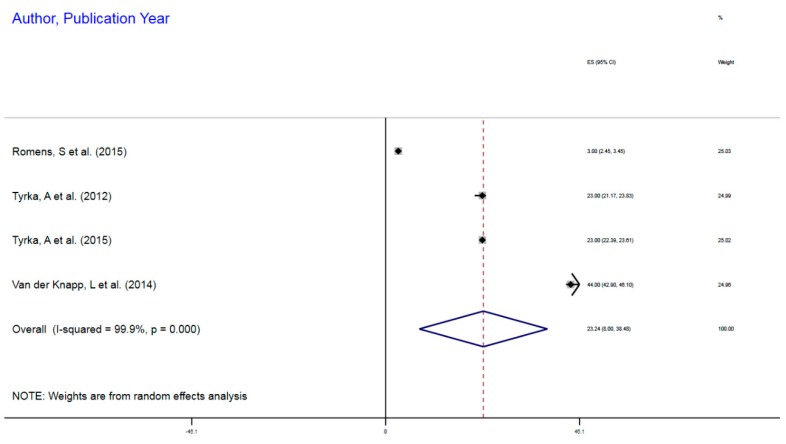
DNA Methylation of NR3C1 (glucocorticoid receptor gene) in Social Adversity with Early Life Stress

**Figure 5 ijerph-16-04280-f005:**
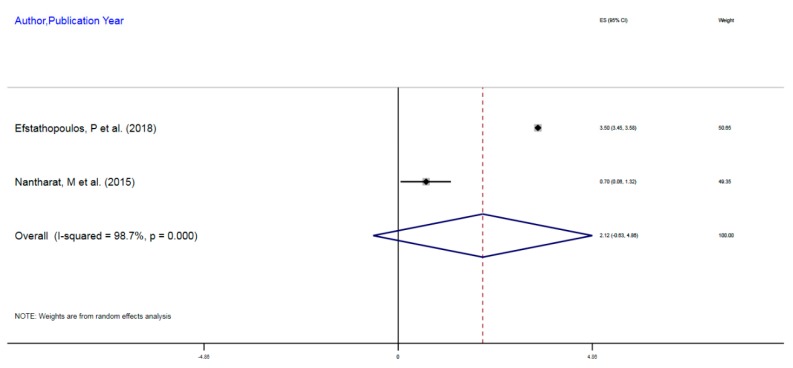
DNA Methylation of NR3C1 (glucocorticoid receptor gene) in Major Depressive Disorder (Unipolar Affective Disorder)

**Table 1 ijerph-16-04280-t001:** mDNA of NR3C1 in Social Adversity and Major Depressive Disorder Correlation.

Author, Year	Exposure	Outcomes	Sample Size	Epigenetic Mechanism	Gene	Gene Region	% Change (MI)	95%CI
Farrell et al. (2011)	Childhood trauma	MDD	77	mDNA	NR3C1	1F Exon Promoter	6%	5.47–6.43
McGowan, P et al. (2009)	Childhood trauma	MDD	36	mDNA	NR3CI	1F Exon Promoter	27%	24.55–31.90
Melas, P et al. (2009)	Childhood sexual abuse	MDD	176	mDNA	NR3C1	1F Exon Promoter	5.4%	5.07–5.73
Perroud, N et al. (2011)	Childhood trauma	MDD	200	mDNA	NR3C1	1F Exon Promoter	1.1%	1.09–1.34
Yehuda, R (2014)	ELS due to parental PTSD	MDD	42	mDNA	NR3C1	1F Exon Promoter	37%	35.4–38.66

Notes and Abbreviations: CI = Confidence Interval; NR3C1= nuclear receptor subfamily 3, group C, member 1, which is a glucocorticoid receptor gene involved in cortisol utilization; 1F Exon = promoter region of the NR3C1 gene; MDD = major depressive disorder; PTSD = post-traumatic stress disorder; ELS = early life stress; mDNA = DNA methylation; MI = Methylation index.

**Table 2 ijerph-16-04280-t002:** DNA Methylation of NR3C1 in Social Adversity.

Author, Year	Exposure	Sample Size	Epigenetic Mechanism	Gene	Gene Region	% Change (MI)	95%CI
Romens, S et al. (2015)	Childhood trauma	56	mDNA	NR3C1	1F Exon Promoter	3%	2.45–3.45
Tyrka, A et al. (2012)	Childhood trauma	99	mDNA	NR3C1	1F Exon Promoter	23%	21.17–23.83
Tyrka, A et al. (2015)	Childhood sexual abuse	184	mDNA	NR3C1	1F Exon Promoter	23%	22.39–23.61
van der Knapp, L et al. (2014)	Childhood trauma	22	mDNA	NR3C1	1F Exon Promoter	44%	42.9–46.1

Notes and Abbreviations: CI = Confidence Interval; NR3C1 = nuclear receptor subfamily 3, group C, member 1, which is a glucocorticoid receptor gene involved in cortisol utilization; 1F Exon = promoter region of the NR3C1 gene; MDD = major depressive disorder; PTSD = post-traumatic stress disorder; ELS = early life stress; mDNA = DNA methylation; MI=Methylation Index.

**Table 3 ijerph-16-04280-t003:** DNA Methylation of NR3C1 in Major Depressive Disorder.

Author, Year	Exposure	Sample Size	Epigenetic Mechanism	Gene	Region on Gene	% Change (MI)	95%CI
Efstathopoulos, P et al. (2018)	Adult MDD	580	mDNA	NR3C1	1F Exon Promoter	3.5%	3.45–3.55
Nantharat, M et al. (2015)	Adult MDD	62	mDNA	NR3CI	1F Exon Promoter	0.7%	0.06–1.32

Notes and Abbreviations: CI = Confidence Interval; NR3C1 = nuclear receptor subfamily 3, group C, member 1, which is a glucocorticoid receptor gene involved in cortisol utilization; 1F Exon = promoter region of the NR3C1 gene; MDD = major depressive disorder; PTSD = post-traumatic stress disorder; ELS = early life stress; mDNA = DNA methylation; Methylation Index.

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
