# Peer review of "Aberrant Epigenomic Modulation of Glucocorticoid Receptor Gene (NR3C1) in Early Life Stress and Major Depressive Disorder Correlation: Systematic Review and Quantitative Evidence Synthesis"

_ijerph, 2019, doi:10.3390/ijerph16214280_

Round 1

Reviewer 1 Report

This paper presents results from a sophisticated meta-analysis of aberrant epigenomic modulation of glucocorticoid receptor gene (NR3C1) in early life stress and major depressive disorder correlation. To date, scientific data for intervention mapping in reducing social adversity and improving human health is still required. Because of this, the authors sought to synthesize the existing data, taking advantage of small, medium, and large sized human studies to carry out a quantitative evidence synthesis (type of meta-analysis).

The methodology seems sound, though I do not consider myself an expert on meta-analyses. However, I do feel that the presentation of these results need some significant clarification prior to publication. I found myself getting confused by the differing terminology throughout the paper, as well as the inconsistencies in the abstract and Figure 1. Could the authors please carefully review their paper to ensure consistency in terminology and consistency in counts? Additionally, there are quite a few careless errors, such as presentation of figures out of order, multiple typographical errors, and excessively small font in the figures. All of this leads to reader confusion and really detracts from an otherwise well done study. I provide examples below:

In the abstract, there are three main analyses described. They include synthesizing the findings of studies involving: Epigenomic modulations (DNAm) following an ELS, resulting in MDD Epigenomic modifications associated with ELS Epigenomic alterations associated with MDD

I think the term “modulations” is appropriate in (a), but using two different terms, “modifications” and “alterations” is a little confusing, unless indeed the authors purport that environmental exposures result in smaller changes (modifications) while a disease process results in stronger changes (alterations) – since modifications are typically considered less than alterations. If that IS the case, then could the authors please use that terminology throughout the paper when referring to the specific objectives? Later in the abstract, they refer to 4 studies assessed for epigenomic “modulation” (as opposed to modifications) and ELS and 2 studies assessed for epigenomic “modifications” (as opposed to alterations). Consistent terminology needs to be presented throughout.

The abstract states, “Of the 592 studies identified through electronic search and screened, 11 met eligibility criteria for inclusion…..” (paraphrasing, 5 for ELS-MDD, 4 for ELS, 2 for MDD). Later, however, in Figure 1, the number 592 is never presented. Instead, there is a starting point of 1,534. Additionally, there are only two branches off this figure, one for the ELS-MDD assessment (finally 5 papers), and one for the ELS assessment (finally 4 papers), but there is not one for the MDD assessment (finally 2 papers). This is confusing and needs to be rectified.

On the same topic, in the first paragraph of the Results, the 2nd sentence is confusing, as it refers to 5 studies meeting criteria for QES, while it was really 11. It seems the authors skipped mentioning the 5 used for ELS-MDD assessment. I think they somehow combined the ideas of the 11 overall studies and the 5 used specifically for the ELS-MDD correlation analysis, as the remainder of the 2nd sentence is not specific to that analysis, but rather general.

The title of Table 1 illustrates another mixing of terminology, “childhood social adversity” and “early life stress.” The authors need to take care to use these terms consistently. Presumably “childhood social adversity” is a subset of ELS, and that needs to be clearly stated and then the terminology used appropriately throughout. Figure 2, which is referring to the same relationship uses the term, “early life stress.” It is not only here that this happens, it also happens on line 369 and the title of Table 2.

Figure 2 is quite impossible to read, since the font is so small. Please fix.

Line 359, should be DNA (not “DN”).

Could the authors clarify what they mean by the “subpopulation DNA methylation”? (Line 360)

The figures jump from Figure 2 to Figure 5. Please fix.

There is a typo in Figure 5 (should be CES, not ES in the column heading).

Please check the rest of the paper, in particular, the Discussion for consistent use of terminology throughout.

Author Response

Dear Editor,

Thank you and the reviewers for the opportunity to revise our manuscript for resubmission for publication in for journal. Below, please find the authors’ response:

Reviewer’s Comment:

This paper presents results from a sophisticated meta-analysis of aberrant epigenomic modulation of glucocorticoid receptor gene (NR3C1) in early life stress and major depressive disorder correlation. To date, scientific data for intervention mapping in reducing social adversity and improving human health is still required. Because of this, the authors sought to synthesize the existing data, taking advantage of small, medium, and large sized human studies to carry out a quantitative evidence synthesis (type of meta-analysis).

Author’s Response:

Thank you very much for your comments. We have realized the relevance of this paper in transforming our approach to public health and in understanding several factors involved in disease process; such as gene and environment interaction.

Reviewer’s Comment:

The methodology seems sound, though I do not consider myself an expert on meta-analyses. However, I do feel that the presentation of these results need some significant clarification prior to publication. I found myself getting confused by the differing terminology throughout the paper, as well as the inconsistencies in the abstract and Figure 1. Could the authors please carefully review their paper to ensure consistency in terminology and consistency in counts? Additionally, there are quite a few careless errors, such as presentation of figures out of order, multiple typographical errors, and excessively small font in the figures. All of this leads to reader confusion and really detracts from an otherwise well done study. I provide examples below:

The title of Table 1 illustrates another mixing of terminology, “childhood social adversity” and “early life stress.” The authors need to take care to use these terms consistently. Presumably “childhood social adversity” is a subset of ELS, and that needs to be clearly stated and then the terminology used appropriately throughout. Figure 2, which is referring to the same relationship uses the term, “early life stress.” It is not only here that this happens, it also happens on line 369 and the title of Table 2.

In the abstract, there are three main analyses described. They include synthesizing the findings of studies involving: Epigenomic modulations (DNAm) following an ELS, resulting in MDD Epigenomic modifications associated with ELS Epigenomic alterations associated with MDD

I think the term “modulations” is appropriate in (a), but using two different terms, “modifications” and “alterations” is a little confusing, unless indeed the authors purport that environmental exposures result in smaller changes (modifications) while a disease process results in stronger changes (alterations) – since modifications are typically considered less than alterations. If that IS the case, then could the authors please use that terminology throughout the paper when referring to the specific objectives? Later in the abstract, they refer to 4 studies assessed for epigenomic “modulation” (as opposed to modifications) and ELS and 2 studies assessed for epigenomic “modifications” (as opposed to alterations). Consistent terminology needs to be presented throughout.

Author’s Response:

Thank you very much for this thoughtful comment. Applied meta-analysis is a process of quantifying scientific studies in order to arrive at an evidence that is generalizable, given the current problem in medicine or public health. As a classic example, we conducted rigorous literature search and identified the flaws in the published literature prior to synthesizing the findings in applied meta-analysis. Despite the fact that, you are not an expert in meta-analysis, your comments and suggestion reflect an individual with a knowledge of systematic review of literature and the potential for us in the presentation of such literature to the scientific community.

We have carefully reviewed the paper and rectified the terminology for consistency. In addition, we have performed a complete readability test to ensure that typos are eliminated prior to the resubmission. Further we have addressed the font in the figures in order to allow for readability and clarity. Furthermore, we have changed the order of the figures by making figure 5, figure 3 and changing the process throughout the paper.  

We also appreciate your comment regarding the interchangeability of the terms modulation changes and alteration, as well as modification. In the scientific parlance, epigenomic changes reflects modulation. However, because there are various mechanistic process in epigenomics the term modification is also interchangeable with modulation and refers more to histone protein acetylation. In effect, because of the perceived distraction or confusion, we have decided to use modulation throughout the entire paper.

Reviewer’s Comment:

The abstract states, “Of the 592 studies identified through electronic search and screened, 11 met eligibility criteria for inclusion…..” (paraphrasing, 5 for ELS-MDD, 4 for ELS, 2 for MDD). Later, however, in Figure 1, the number 592 is never presented. Instead, there is a starting point of 1,534. Additionally, there are only two branches off this figure, one for the ELS-MDD assessment (finally 5 papers), and one for the ELS assessment (finally 4 papers), but there is not one for the MDD assessment (finally 2 papers). This is confusing and needs to be rectified.

Author’s Response:

We are very appreciative of this comment and observation. The flow chart began with 1,534 which is basically the articles identified. However, there were 592 screened articles as reflected on the flow chart, implying 386 and 206. The flow chart had been modified to reflect the 592 studies utilized as well as the 2 studies on MDD and DNA methylation.

Reviewer’s Comment:

On the same topic, in the first paragraph of the Results, the 2nd sentence is confusing, as it refers to 5 studies meeting criteria for QES, while it was really 11. It seems the authors skipped mentioning the 5 used for ELS-MDD assessment. I think they somehow combined the ideas of the 11 overall studies and the 5 used specifically for the ELS-MDD correlation analysis, as the remainder of the 2nd sentence is not specific to that analysis, but rather general.

Author’s Response:

Thank you very much for the observation and comments. We have reviewed the results section and have addressed this issue. The paragraph is very clear by identifying the two studies associated with MDD only and 5 studies associated with ELS and MDD, as well as the 4 studies associated with ELS only.

Reviewer’s Comment:

Figure 2 is quite impossible to read, since the font is so small. Please fix.

Author’s Response:

We have decided to change the resolution of the figures and currently the figures are readable.

Reviewer’s Comment:

Line 359, should be DNA (not “DN”).

Author’s Response:

We addressed this by adding the A to the DN implying DNA methylation.

Reviewer’s Comment:

Could the authors clarify what they mean by the “subpopulation DNA methylation”? (Line 360)

Author’s Response:

In meta-analysis, even though you are not an expert, the subpopulation refer to meta-regression, implying that within a group there are point estimates, or summary estimates, of individual groups which allows for the term meta-regression. We have changed the word subpopulation to meta-regression, which is clearly what is understood in meta-analysis. Specifically, the diamonds in Figure 5, which is now changed to Figure 3, reflects the subpopulations based on the methylation profile.

Reviewer’s Comment:

The figures jump from Figure 2 to Figure 5. Please fix.

Author’s Response:

Thank you very much for this observation. We have addressed this by citing Figure 5 as Figure 3.

Reviewer’s Comment:

There is a typo in Figure 5 (should be CES, not ES in the column heading).

Author’s Response:

Thank you very much for this observation. The ES refers to effect size. However, because it is a standardized term in meta-analysis we are unable to transform this in the column. The application of CES is based on the researchers understanding of the summery point estimate as the common effect size (CES) implying the combination of the individual effect size’s in order to generate the comment effect size.

Reviewer’s Comment:

Please check the rest of the paper, in particular, the Discussion for consistent use of terminology throughout. 

Author’s Response:

Thank you very much for this observation and comments. We have revised the paper three times and have identified some of the inconsistencies, addressed the typos, and improved the abbreviations, as well as the overall clarity of the paper.

Thank you for your remarkable comments and suggestions with the hope that these findings will be disseminated very soon to the scientific community.

Reviewer 2 Report

Great review regarding the effects of stress on epigenetic modifications in the GR gene and the correlation with aberrant changes in gene expression. Very relevant topic and very good review of the literature. I recommend acceptance of this manuscript.

One minor issue: There is an excessive use of abbreviations. The authors may consider spelling out the terms to facilitate understanding and reading of the manuscript. 

Author Response

Reviewer's Comment:

Great review regarding the effects of stress on epigenetic modifications in the GR gene and the correlation with aberrant changes in gene expression. Very relevant topic and very good review of the literature. I recommend acceptance of this manuscript.

One minor issue: There is an excessive use of abbreviations. The authors may consider spelling out the terms to facilitate understanding and reading of the manuscript.

Author’s Response:

Thank you very much for the recognition provided to this manuscript and your recommendation for its acceptability. Also, we really appreciate your comment regarding the abbreviations. Epigenomic findings can be very challenging and without careful consideration readers may be confused. We have addressed these abbreviations in the paper with the hope that this will increase its readability.